# A New Role of Acute Phase Proteins: Local Production Is an Ancient, General Stress-Response System of Mammalian Cells

**DOI:** 10.3390/ijms23062972

**Published:** 2022-03-10

**Authors:** Péter Hamar

**Affiliations:** Institute of Translational Medicine, Semmelweis University Budapest, Tüzoltó u 37-47, 1094 Budapest, Hungary; hamar.peter@med.semmelweis-univ.hu; Tel.: +36-20-825-9751

**Keywords:** acute-phase protein local production, APP, cancer, kidney fibrosis, ancient stress response

## Abstract

The prevailing general view of acute-phase proteins (APPs) is that they are produced by the liver in response to the stress of the body as part of a systemic acute-phase response. We demonstrated a coordinated, local production of these proteins upon cell stress by the stressed cells. The local, stress-induced APP production has been demonstrated in different tissues (kidney, breast cancer) and with different stressors (hypoxia, fibrosis and electromagnetic heat). Thus, this local acute-phase response (APR) seems to be a universal mechanism. APP production is an ancient defense mechanism observed in nematodes and fruit flies as well. Local APP production at the tissue level is also supported by sporadic literature data for single proteins; however, the complex, coordinated, local appearance of this stress response has been first demonstrated only recently. Although a number of literature data are available for the local production of single acute-phase proteins, their interpretation as a local, coordinated stress response is new. A better understanding of the role of APPs in cellular stress response may also be of diagnostic/prognostic and therapeutic significance.

## 1. Introduction

During the stress response of the body, the liver produces acute-phase proteins (APPs) [1]. The known effect of stress on cells/organs is the heat shock response. We have demonstrated a general stress response of acute-phase proteins, termed the acute-phase response (APR), as the main proteomic response in different mouse models including acute kidney injury (AKI)-induced renal fibrosis and modulated electro-hyperthermia (mEHT)-induced death of triple-negative breast cancer (TNBC) by multiplex methods. We observed that during renal scarring due to severe ischemic damage to the kidney and during heat therapy treatment of mouse breast cancer, the kidney and the tumor cells produce acute-phase proteins, and this is the leading response of these cells according to multiplex studies.

There is scarce information in the literature that APR proteins are produced outside of the liver and locally, in stressed cells. The APR was seen as independent of the model and tissue. Based on these observations, we propose that the APR is a coordinated, ancient defense mechanism of different cells similar to the heat shock response. This hypothesis is supported by sporadic literature data, although these publications examined the role of a single or a few acute-phase proteins.

Better characterization of the local APR may lead to better diagnostic/prognostic or therapeutic possibilities at least in the fields of AKI-induced fibrosis and cancer. Possible clinical applications of the APR theory are: (1) Inhibition of the tumor stress response can increase the efficacy of various tumor treatment procedures. (2) Influencing the same stress response may slow renal scarring following ischemia. (3) The detection of stress proteins and their degradation products in the bloodstream and urine may be an effective new tool for clinical monitoring of these processes.

## 2. Main

### 2.1. Common Pathways of Renal Fibrosis and Cancer Progression

Renal fibrosis and cancer progression have been linked through several common pathomechanistic pathways including tissue hypoxia, inflammation and oxidative stress. Furthermore, there are several common mediators, such as transforming growth factor beta (TGF-beta)—a key driver of malignant transformation—as well as tissue fibrosis, the involvement of fibroblasts in fibrosis and in the tumor microenvironment, or hypoxia-inducible factor-1 (HIF-1) [2,3,4,5]. These factors (TGF-beta fibroblasts, HIF-1alpha) are the most important factors in wound healing and tissue repair; however, if they remain activated following healing of an acute injury such as hypoxia, they are the key factors of fibrosis progression. Furthermore, these factors are also regarded as important tumor promoters.

### 2.2. The Heat Shock Response and the Acute Phase Response

The heat shock response (HSR) is a well-described, general and ancient stress response of cells to several different types of stress [6,7,8]. The acute-phase response (APR) is part of a general, systemic response to infections and tissue damage. According to the definition, proteins whose plasma concentration is changed by at least 25 percent in response to pro-inflammatory stimuli are termed acute-phase proteins (APPs) [9]. They have a role in restoring homeostasis after inflammation [10]. The generally held concept is that APPs are produced in the liver and triggered mainly by inflammatory interleukin-6 (IL-6) (as well as IL-1, IL-8 and TNF-alpha) and secreted into the blood. However, APPs are also synthesized in other organs. Thus, APPs can contribute to local defense responses and repair mechanisms [11]. We have demonstrated by proteomic analysis a complex local production of several APPs in different tissues upon injury, such as electromagnetic heating of tumor cells [12], or acute hypoxic [13] or chronic fibrotic injury of the kidney [14]. In all of these studies [6,12,13,14], >50 percent of the top 20 upregulated proteins were APPs, as detected by next-generation sequencing (NGS) and verified by nanostring, PCR or mass spectrometry (MS). Thus, we propose that local APP production may be a similar, organized, ancient stress response to the well-known HSR. As the HSR serves to protect cells from stress, the local APR may have a similar protective role. Inhibition of the HSR can support cancer therapies as demonstrated by us [6] and others. We propose similar roles for the local APR to the well-known local HSR.

### 2.3. Breast Cancer

Breast cancer was the most common cancer among women worldwide, contributing 25.8 percent of the total number of new cases diagnosed in 2020 [15]. Triple-negative breast cancer (TNBC) is a highly aggressive breast cancer type with very poor survival due to the lack of targeted therapy [16]. We utilize the mouse TNBC model: implantation of isogenic cell lines derived from a spontaneous tumor in Balb/c mice (410.4). Isolated subclones vary in aggressivity and metastatic potential. The most aggressive and invasive subclones are the 4T1 and 4T07 lines (Figure 1A) [17]. The model enables to study immune mechanisms in TNBC [6].

### 2.4. Modulated Electro-Hyperthermia (mEHT)

Modulated electro-hyperthermia (mEHT) is a newly emerging adjuvant cancer treatment used in human oncology. During mEHT, a focused electromagnetic field (EMF) is generated within the tumor by applying capacitive radiofrequency. Selective energy absorption by the tumor is the consequence of elevated oxidative glycolysis (Warburg effect) and conductivity of the tumor. The EMF induces cell death by thermal and non-thermal effects. Capacitive energy delivery and frequency modulation enable the application of non-thermal effects. The company inventing and producing mEHT devices for human therapy has developed the rodent mEHT device to enable the accurate, reproducible, standard and effective treatment of TNBC in the inguinal region of mice [6] (Figure 1B upper panel). Selective energy absorption enables +2.5 C heating of the tumor (Figure 1B lower panel). Thus, the thermal and non-thermal effects amplify each other, leading to effective tumor-cell killing. The device is also useful to deliver therapeutic agents locally to the tumor or the kidney by thermo-sensitive liposomes (TSL).

### 2.5. Acute (AKI) and Chronic Kidney Disease (CKD)

Both acute (AKI) and chronic kidney disease (CKD) that can result in renal failure are common, significant, but underestimated problems regarding their relevance. CKD is becoming a major health care problem around the world for the aging population and due to the growing incidence of hypertension. AKI is a devastating common disease that can heal or be cured in some cases, but the acute injury can initiate unfavorable processes culminating in renal failure. Better understanding the AKI-induced CKD may lead to new therapeutic avenues for halting the progression of the presently incurable renal fibrosis. Following damage of tubular epithelial cells (TEC), the development of the highly differentiated form of TEC can be disturbed during the regenerative processes leading to pathological changes of intrinsic regulatory processes. Several factors, such as TGF-beta [6,12,13,14], miR-21, HIF-1alpha, etc., are considered protective and stimulate regenerative processes following acute tubular injury, but sustained elevation of these factors is the major driving force of chronic scarring and fibrosis (CKD) leading to eventual renal failure. We investigated AKI and CKD in three mouse models: endotoxin (LPS)-induced renal ischemia [13], renal ischemia-reperfusion-induced fibrosis [18] and miR-193 transgene-induced fibrosis [14].

In miR-193 transgenic mice, progressive glomerular sclerosis (GS) leads to renal fibrosis and failure [19]. During the course of this GS, we have observed the progressive accumulation of a hyaline-like material within the glomerular extracellular matrix (ECM) (Figure 2A). Mass spectrometric (MS) analysis of this scarring ECM demonstrated that all three fibrinogen chains were within the top 10 most abundant proteins. The hyaline-like accumulating material within the glomerular ECM stained positive for fibrinogen (Figure 2B). Besides fibrinogens, several other APPs and complement-related proteins were abundant in the glomerular ECM as detected by MS [14].

The most exciting model of renal ischemia-reperfusion injury-induced fibrosis is the result of our recent demonstration that fibrosis progression is halted by delayed removal of the contralateral kidney. Fibrosis is accelerated if the non-injured kidney is left in place and leads to a small, kidney-shaped scar tissue within 4 weeks (Figure 2C). However, removal of the non-injured kidney slows down the progression of fibrosis dramatically, and the post-ischemic kidneys are still functional 4 months after the initial ischemia-reperfusion injury [18].

### 2.6. The Potential of Degradomic Analysis

Pathological states such as cancer, ischemia and inflammation can affect various proteolytic pathways, which usually lead to increased general proteolysis within the organism. Identification of the affected proteolytic substrates can significantly improve our understanding of specific pathological mechanisms. Furthermore, since proteolytic peptides and protein fragments diffuse from the primary tissues to body fluids, such as blood and urine, such fluids can serve as good sources for protease substrate identification and their utilization as diagnostic, prognostic or therapeutic biomarkers. Thus, the effect of physiological stress on general proteolysis can be determined by performing the peptidomic analysis of urine and/or blood samples collected from subjects with cancer or postischemic renal fibrosis. APPs have been proposed already as potential biomarkers of cancer and renal ischemia/fibrosis [11].

Ischemia-related stress is known to cause various proteolytic events, and monitoring of cleaved protease substrates can provide information on stress-related molecular mechanisms, which can have a significant prognostic, diagnostic, and therapeutic value [20,21]. The upregulation of protease expression and proteolytic activity is implicated also in numerous pathological conditions, such as cancer, cardiovascular and other diseases [22,23]. During disease progression, proteases produce characteristic patterns of cleaved proteins, which characterize severity and progression. As proteolytic fragments and peptides generated in the affected tissue are commonly translocated to body fluids, such as blood and urine, their possible application as biomarkers can be done by degradomic analysis [24]. The urine is an especially suitable, non-invasive sample source in the case of renal diseases. Thus, measuring degraded products of APPs produced by cancer and during postischemic renal fibrosis in blood and urine samples of mice has a great diagnostic/prognostic potential. Indeed, numerous proteolytic products and cleavage events in various biological contexts have been already identified [23,24,25].

## 3. Previous Findings

Multiplex analysis of all of the above models resulted in the identification of a proteomic response lead by the above-detailed APPs. In a comparison of the raw data from our four papers [6,12,13,14], analyzing tissue/cell stress with multiplex methods revealed that several APPs were upregulated in more than one model. Furthermore, real-time polymerase chain reaction (RT-PCR) detection of messenger RNA (mRNA) and cell culture studies verified that APPs were produced by the tumor cells or the renal tissue, suggesting that APPs are produced by the stressed tumor/renal cells and are not originating from the liver. Indeed, several previous publications support our observation. Increased renal production of fibrinogen, ceruloplasmin, complement C3, haptoglobin, hemopexin, serum amyloid A, beta-2-microglobulin, alpha-1-acid glycoprotein and plasminogen activator inhibitor-1 have been described. Connections between chronic inflammation, the APR and glomerular fibrin deposition have been observed by others (Harald Mischak, Professor of Proteomics, Univ. Galsgow, Mosaiq Diagnostics—personal communication). They have demonstrated an interplay between inflammation (APR) and fibrinolysis in ischemic heart disease. Although these scarce and isolated findings demonstrate the renal production of some APPs in sepsis, the coordinated, complex APR of the kidney has not been described before. Similar findings demonstrating local production of APPs by cancer cells have been demonstrated (see later). A proteomic analysis of the urine revealed APPs (FGA, FGG, HP, ITI4, SERPINA1) [20] as biomarkers for prostate cancer, but the paper lacks the analysis of their pathophysiological role.

## 4. Known Roles of Major Acute-Phase Proteins Detected by NGS in Different Models

### 4.1. Fibrinogens

Fibrin(ogen) (FN) is an abundant protein, present in human blood at concentrations of 1.5–4 g/L [26]. The physiological functions of FN are: besides providing a scaffold for blood clotting, FN is important for the assembly of (extracellular) matrices to enhance host defense [27]. FN has been implicated in pathological processes including renal diseases and cancer. Fg-deficient mice (Fg ^−/−^) [28] were protected from endotoxemia [29], renal fibrosis [30] and ischemia-reperfusion injury [31]. Coagulation factors have been linked with malignancy for over 100 years and high plasma fibrinogen, in particular, has been associated with cancer progression. FN can be produced by cancer cells, such as breast cancer [27], and binds to and surrounds cancer cells, forming a protective structure (Figure 3A). Furthermore, by interacting with endothelial cells, FN contributes to the extravasation of cancer cells [26]. A hallmark of breast carcinoma is the local synthesis and deposition of fibrinogen (precipitation without conversion to fibrin) [32]. FN in the ECM augments the innate immune response to tissue injury or cancer [32]. FN deposition is a predominant component in breast tumor stroma [32]. All three FN chains were within the top upregulated genes on all three multiplex screens in TNBC and renal models. Our multiplex data do not distinguish between fibrinogen and fibrin and do not indicate the involvement of thrombin or other coagulation-related factors. Besides FN upregulation, fibrinolysis inhibitors Serpin A3 (alpha-1-Antichymotrypsin) and alpha-2-macroglobulin were also upregulated on both the tumor and at least one renal screen (Figure 3B).

Thus, modulating FN production may have a therapeutic potential. Our mouse breast cancer model of sibling cell lines with different metastatic potential (Figure 1D) is perfect to study the concept of FN’s role in tumor progression and metastasis. Proof-of-concept experiments will be carried out in FN-deficient mice and cell lines. As a potential therapeutic intervention, precipitated fibrin can be removed from the tumor microenvironment by treatment with plasmin activators (Alteplase or Batroxobin) (Figure 3B). Similar experiments in FN knockout mice (proof-of-concept) and treatment with Alteplase (to dissolve precipitated FN) are planned in the renal ischemia-induced fibrosis mouse model. The role of FN in cancer progression and renal fibrosis will be investigated first by using fibrinogen knockout mouse breast cancer cell lines and in fibrinogen KO mice.

### 4.2. Haptoglobin

Haptoglobin (Hp) is a well-characterized glycoprotein found in all mammals but also, in its simpler form, in fish. However, its exact physiological function is still uncertain [37]. Hp functions as an antioxidant to prevent oxidative damage to tissues, including the kidney [38]. Further, already-described Hp functions include pro-angiogenic [39] and immunomodulatory [40] functions through inhibition of cathepsin-B [41] and prostaglandin synthase [37]. Moreover, especially haptoglobin but also fibrinogen beta were indicative of metastasis in a human triple-negative MDA-MB-231 breast cancer model [42].

Local tissue production of haptoglobin (Hp) including the kidney is not negli-gible [38,43]. Under pathological circumstances, Hp was locally expressed in oncological tissues [44]. Hp production was also induced by ischemia in tubular epithelial cells of the kidney after an ischemic injury [45] as a stress response. Six different forms of acute kidney injury (AKI) including ischemia-reperfusion injury and endotoxin shock evoked rapid, striking, and sustained induction of the proximal tubule Hp gene, leading to around 10- to 100-fold renal Hp protein elevations. Endotoxemia evoked 25-fold greater Hp mRNA increases in the kidney than in the liver [45]. Haptoglobin has been demonstrated to be produced in comparable levels to the liver in several cancer models, such as non-small cell lung cancer (NSCLC) [46], colorectal cancer (CRC) [47], endothelial adenocarcinoma [48] and ovarian cancer [49]. In some of these studies, elevated local Hp production [49] or systemic Hp level [50] was associated with poor prognosis. Haptoglobin was several fold upregulated on the NGS, nanostring and MS analysis of mEHT-treated breast cancer and on the MS screen validated by PCR in the ischemic kidneys. Further genes/proteins involved in the APR and iron homeostasis (hemopexin, transferrin, ferritin) were significantly upregulated in both the mEHT-treated tumors and the ischemic kidneys, supporting a general activation of this group of proteins by cell/tissue injury.

### 4.3. Protease Inhibitors (Serpins, ITI)

Proteases are involved in many, if not all, acute diseases and are found in all living organisms, including plants [51]. Within the serine protease group, the chymotrypsin family includes numerous proteases within the complement system, the contact activation (kallikrein)/coagulation system, and the fibrinolytic system [51].

#### 4.3.1. Serpins

Serine protease inhibitors (SERPINs) are an evolutionary old, structurally conserved [52] superfamily consisting of at least 37 proteins divided into 16 clades (A-P) [53] in humans. They regulate coagulation, inflammation and wound healing by inhibiting protease activity or function as chaperones [54], and their dysregulation is associated with pathologies including inflammation and cancer [55,56,57]. In our studies, several serpin-clade A members—1, 1A, 1C (alpha-1-antitrypsin), 3B (alpha-1-antichymotrypsin), 3C, 3K (kallikrein inhibitor), 3M, 3N, and 10 (protein Z-dependent protease inhibitor)—were elevated at least two-fold in at least one study. A1 and A3K were elevated in both of our nephrology studies, whereas 3N was elevated both in the mEHT-treated tumors and in ischemic kidneys. The member 3 of clade A (SerpinA-3), also known as alpha-1-antichymotrypsin, inhibits several proteases (chymase, chymotrypsin, kallikreins and cathepsin G) and thus limits inflammation, coagulation, ECM remodeling and apoptosis and has been implicated in tissue fibrosis [58]. SerpinA3 is also expressed by carcinomas regulating apoptosis and invasiveness [59,60]. SeprinA-3N was demonstrated by others to be upregulated in the kidney 60-fold 2 days after sepsis-induced renal hypoxia [61] in a proteomic study. Serpina3N produced in tubular epithelial cells was elevated in the urine of rats with early AKI-to-CKD transition [62]. Furthermore, SerpinA3 overexpression decreased cell adhesion to the extracellular matrix and to neighboring cells and protected them from apoptosis [58]. SerpinA-3N was upregulated upon injury [54], including hypoxia [58], and could serve as a marker of the transition of AKI to CKD [62]. Thus, especially SerpinA-3N seems to be a ubiquitous stress-response protein involved both in cancer protection from therapy and renal ischemia as well as AKI-to-CKD transition in the kidney.

#### 4.3.2. Inter-Alpha-Trypsin Inhibitor (ITI)

Inter-alpha-trypsin inhibitor (IalphaI/ITI) family members are ancient molecules that evolved over a hundred million years. IalphaI is a complex containing the proteoglycan bikunin (alpha-1-microglobulin (A1M) / bikunin precursor: AMBP) as a light chain to which one of five types of homologous heavy chains are covalently attached. Its most described roles are scavenging free radicals [63] and ECM protection through (1) inhibition of proteases and complement; (2) interaction with ECM proteins, most notably hyaluronan; and (3) cell regulation. A1M is a regulator of the intracellular redox environment and the ER folding and posttranslational modification processes. ITI inhibits trypsin, plasmin, and lysosomal granulocytic elastase and so has an anti-inflammatory role [64]. Thus, ITI plays a pivotal role in tissue homeostasis [65]. The significance of ITI has been described in inflammatory and malignant diseases [66]. ITIH-3 and -4 heavy chains have a potential diagnostic utility as biomarkers for breast cancer [66] and for ischemic disease [66,67]. In our studies, the ITI light chain (AMBP) was upregulated in mEHT-treated cancer and fibrotic kidney, whereas four heavy chain types (1–4) were upregulated in the cancer model, but only ITIH-1 was upregulated in both the ischemic and the fibrotic kidney models.

### 4.4. Alpha-2-Macroglobulin (a2-MG)

Alpha-2-macroglobulin is a well-known acute-phase protein produced by the liver, but local production by macrophages and fibroblasts has also been described. In our studies, both in mEHT-treated tumors and in hypoxic kidneys, a2-MG was upregulated greater than two-fold both at the mRNA and protein levels. A2-MG is an antiprotease: it inactivates an enormous variety of proteinases. It inhibits fibrinolysis by inhibiting both plasmin and kallikrein.

### 4.5. Complement Factors (C4B)

Several complement components (C2, C3, C4B, C7, C8 alpha and gamma chains), complement regulators (CD59A glycoprotein - MAC inhibitor), complement factors B, D, H, I and P (properidin) and the pentraxin-related gene (PTX3) were upregulated on our multiplex screens. From these, C4B was three-fold (protein-MS) and four-fold (mRNA-NGS, nanostring) upregulated in mEHT-treated tumors and three-fold (mRNA-NGS) in fibrotic kidneys. Furthermore, CFB, D, I and P appeared on these two screens. The complement system is the central effector of the humoral arm of innate immunity: the more ancient part of the immune system. However, both C3 and C4 are essential parts of both the innate and the acquired immune system [68]. C4B is the proteolytically active component of the C3 and C5 convertases. The alternative pathway is triggered by the activation of C4B and acts as an amplification loop of the complement system [69]. A small-molecule factor B inhibitor has been described recently [69].

## 5. Summary and Future Perspectives

In conclusion, the presently prevailing concept of the acute-phase response (APR) may be only part of the truth. Besides the well-known systemic acute-phase response, where in response to tissue injury and consequent inflammation, locally produced inflammatory cytokines (IL-6, IL-1b, etc.) induce acute-phase protein (APP) production by the liver, there is a local APR at the cellular level: a coordinated local production of several acute-phase proteins with the aim of protecting the cells from environmental stress, similarly to the also well-described heat shock response. A future perspective includes the possible utilization of this novel stress response for diagnostic and therapeutic purposes. In the case of cancers, the cancer cells may produce APPs to protect themselves from conventional (chemotherapy, radiotherapy) or adjuvant (heat therapy) treatments. Local, targeted inhibition of the production of these APPs in the tumor cells may substantially enhance the anti-tumor effects of presently avialable anti-cancer treatments.

## Figures and Tables

**Figure 1 ijms-23-02972-f001:**
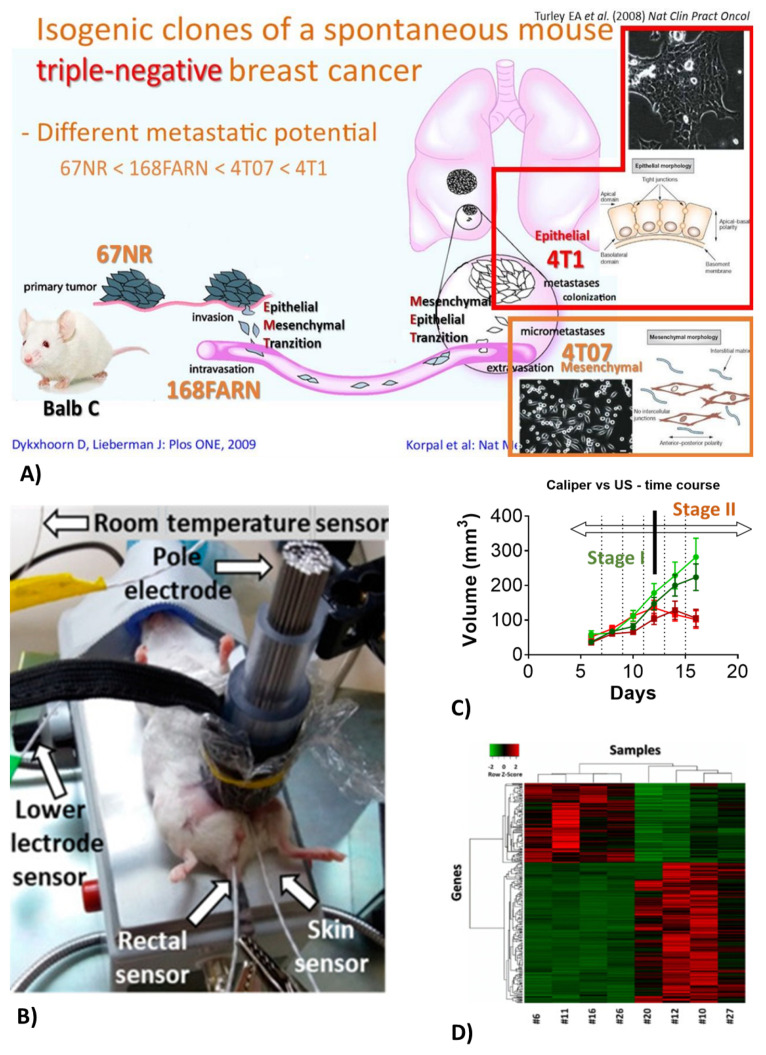
(**A**) Subclones of the mouse mammary carcinoma cell line 410.4 with different metastatic potentials [17]. (**B** upper panel) The rodent-modulated electro-hyperthermia (mEHT) device [6]. (**B** lower panel) Temperatures measured within the tumor are 2.5 °C higher than in the surroundings with very little deviation [6]. (**C**) Tumor size starts to be reduced following at least 3 treatments [12]. (**D**) Multiplex (next-generation sequencing) result of mEHT vs. sham-treated tumors demonstrate clear effects on gene expression [6,12].

**Figure 2 ijms-23-02972-f002:**
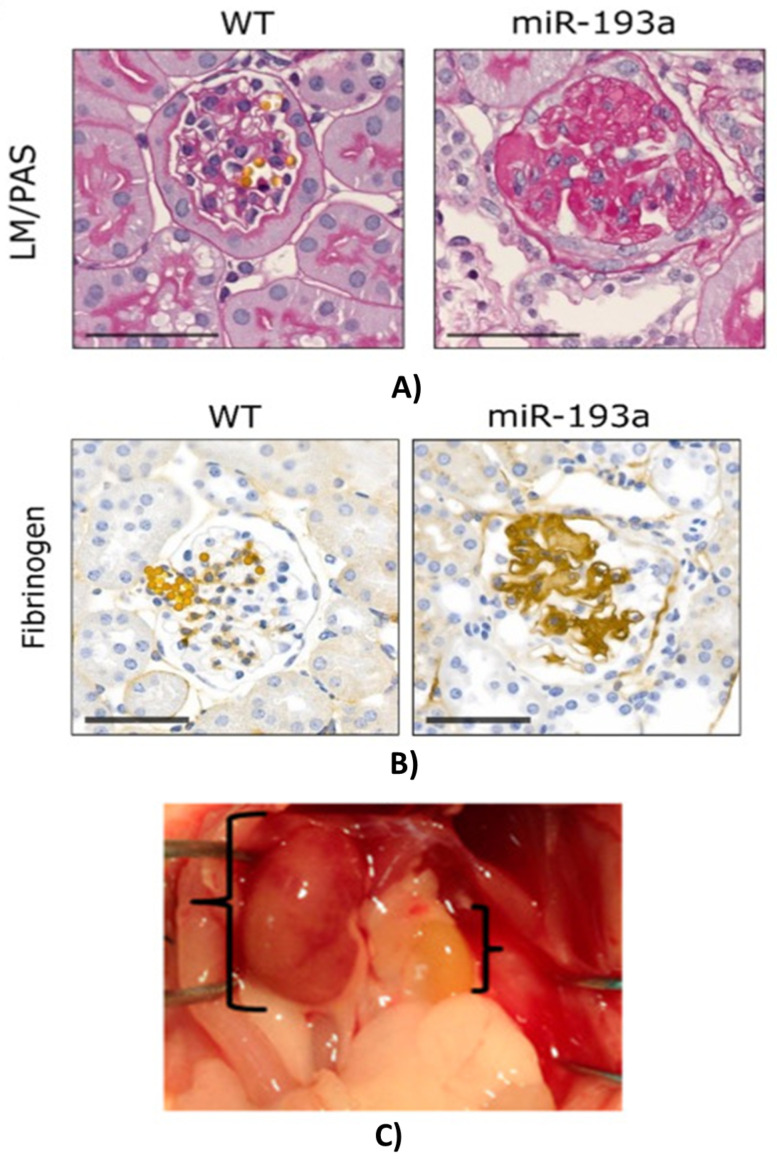
The miR-193 transgenic (**A**,**B**) [14] and ischemia-induced (**C**) [18] models of renal fibrosis. (**A**) Hematoxylin-eosin staining showed hyaline-like deposition within the glomeruli. (**B**) The precipitate was rich in fibrin (immunohistochemistry) [14]. (**C**) Normal (right) and scarring kidney (left) 4 weeks after 30 min of ischemia-reperfusion injury of the left kidney.

**Figure 3 ijms-23-02972-f003:**
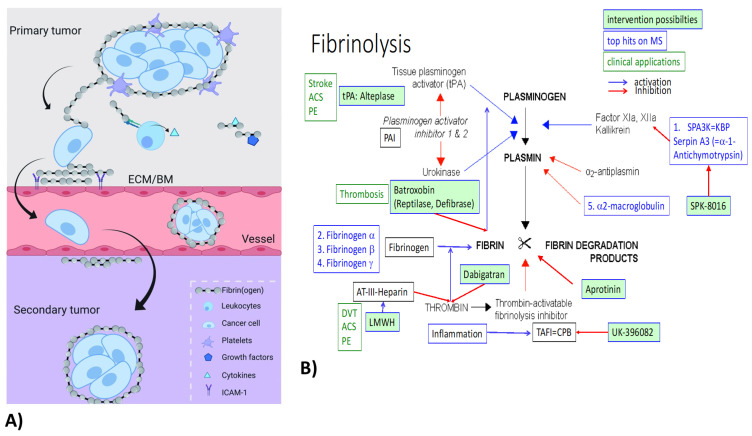
(**A**) Fibrin(ogen) surrounding tumor cells aids the formation of protective tumor microenvironment (TME) and metastasis [26]. (**B**) Fibrinolysis pathways. Top upregulated proteins (blue) in our renal fibrosis mouse model were related to fibrin(ogen) deposition and degradation. (tPA: tissue plasminogen activator [33], PAI: plasminogen activator inhibitor, AT-III: antithrombin-3, TAFI: thrombin activatable fibrinolysis inhibitor, CPB: carboxypeptidase-B, SPA3K: serine protease inhibitor (Serpin)-A3 = alpha-1-anitchymotrypsin.) Current possibilities for intervention (green labels) and diseases where these interventions are used clinically (green letters) are indicated. (ACS: acute coronary syndrome, DVT: deep vein thrombosis, PE: pulmonary embolism, tPA: tissue plasminogen activator, LMWH: low-molecular-weight heparin, Dabigatran [34]: direct thrombin inhibitor (Pradaxa), SPK: Spa3K inhibitor developed by Spark Therapeutics [35], UK-396,082: TAFI inhibitor [36]).

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
