# Peer review of "A New Role of Acute Phase Proteins: Local Production Is an Ancient, General Stress-Response System of Mammalian Cells"

_ijms, 2022, doi:10.3390/ijms23062972_

Round 1

Reviewer 1 Report

The manuscript does not offer particular updates to the topic of acute-phase proteins (APP). The paragraphs, although interesting, do not seem to create new ideas for future research and seem to contain little information, with few references.
In fact, the paper represents a sort of "state of the art" without providing future research topics to the reader, even on the subject of carcinomas.
Furthermore, there are many self-citations (n.9), some of which are not very relevant.

Author Response

Explanation of revision:

1.) 2.3 breast cancer part revised. We rephrased the paragraph describing breast cancer relevance for our research according to the reviewers suggestions. The main message was not changed.

2.) Figure 2/A removed. Reference numbers of original articles added in each figures description where they were needed. Reference numbers, figures references and reference list updated according to theese changes in the text.

About copyrights of figures:

  1. 1/a Danics L. (2018): Modulated electro hyperthermia inhibits tumor progression in a triple negative mouse breast cancer model; Oncothermia Journal 24: 442-454

ICHS conference presentation of our colleague published in Oncothermia Journal http://www.oncothermia-journal.com/browse-volumes-2/page/3/

  1. 1/b/upper panel: Danics, 2020: https://www.ncbi.nlm.nih.gov/pmc/articles/PMC7565562/11/B, 12/D
  2. 1/b lower panel: unpublished/original based on: Danics, 2020: https://www.ncbi.nlm.nih.gov/pmc/articles/PMC7565562/2/A
  3. 1/c Schvarcz, 2021: https://www.ncbi.nlm.nih.gov/pmc/articles/PMC8038813/ / fig. 1/B
  4. 1/d Schvarcz, 2021: https://www.ncbi.nlm.nih.gov/pmc/articles/PMC8038813/ / fig. 5/A
  5. 2/a Bukosza, 2020: https://www.ncbi.nlm.nih.gov/pmc/articles/PMC7139641/ / fig. 1/A
  6. 2/b Bukosza, 2020: https://www.ncbi.nlm.nih.gov/pmc/articles/PMC7139641/ / fig. 2/A
  7. 2/c Tod P, 2020: https://www.ncbi.nlm.nih.gov/pmc/articles/PMC7312122/ / fig.4/A
  8. 3/a Vilar, 2020: https://www.ncbi.nlm.nih.gov/pmc/articles/PMC7012490// fig. 2
  9. 3/b original image

2-8. are cited from our previously published papers in open access journals (OAJ), where the copyright statement is: ’This article is an open access article distributed under the terms and conditions of the Creative Commons Attribution (CC BY) license (https://creativecommons.org/licenses/by/4.0/)’.

For 9. we asked copyright permission from the journal. Please find attached the permission.

Reviewer 2 Report

Dear authors.

The manuscript presented by the authors is a novel topic with great research potential. The introduction of this review is very well focused, as it immediately manifests the topic to be discussed later. The pathological situations mentioned in the separate sections of the paper are very relevant, and acute phase proteins are also important in the acute phase response. However, there is no link between the two sections. I suggest the authors to interlace both sections, updating the description of the state of the art in the local production of APPs in the mentioned pathologies.  I would like to encourage the authors to rethink the flow of ideas of this interesting review, as it is a contribution to research in this area.

In the abstract the abbreviation AFR is mentioned, which was not presented before, nor in the body of the manuscript.

The figures may be a contribution, but in the printed version they are very small and unreadable.

Author Response

(The authors gave the same response as above.)

Reviewer 3 Report

After carefully reading the manuscript, I conclude that the abstract, introduction, and other chapters cover the issues discussed in an extensive and proper manner.

Although it is a valuable work having an interesting idea it needs some adjustment:

  • Since a large number of abbreviations are used along the text, I recommend that all used abbreviations should be listed at the end of the main body of text before references. This will certainly provide the reader with a better understanding of the context of the issues discussed. In addition, this is a kind of standard in contemporary scientific papers.
  • The authors should add section covering the conclusion and future perspective in the form of an additional chapter.

I recommend publication after minor revision.

Author Response

(The authors gave the same response as above.)

Round 2

Reviewer 1 Report

The manuscript has been improved, however the author has included many self-citations by disregarding previous revisions. In my opinion the paper does not provide adequate contributions on the subject.

Reviewer 2 Report

Dear authors.

No more comments. I am satisfied with incorporated changes.

Best regards.